# Wake steering optimization under uncertainty

**Julian Quick[1,2], Jennifer King[2], Ryan N. King[2], Peter E. Hamlington[1], and Katherine Dykes[2]**

[1]University of Colorado, Boulder, CO, USA
[2]National Renewable Energy Laboratory, Golden, CO, USA

**Correspondence:** Julian Quick (julian.quick@colorado.edu)

**Abstract.** Turbines in wind power plants experience significant power losses when wakes from upstream turbines affect the energy production of downstream turbines. A promising plant-level control strategy to reduce these losses is wake steering, where upstream turbines are yawed to direct wakes away from downstream turbines. However, there are significant uncertainties in many aspects of the wake steering problem. For example, infield sensors do not give perfect information, and inflow to the plant is complex and difficult to forecast with available information, even over short time periods. Here, we formulate and solve an optimization under uncertainty (OUU) problem for determining optimal plant-level wake steering strategies in the presence of independent uncertainties in the direction, speed, turbulence intensity, and shear of the incoming wind, as well as in turbine yaw positions. The OUU wake steering strategy is first examined for a two-turbine test case to explore the impacts of different types of inflow uncertainties, and is then demonstrated for a more realistic 11-turbine wind power plant. Of the sources of uncertainty considered, we find that wake steering strategies are most sensitive to uncertainties in the wind speed and direction. When maximizing expected power production, the OUU strategy also tends to favor smaller yaw angles, which have been shown in previous work to reduce turbine loading. Ultimately, the plant-level wake steering strategy formulated using an OUU approach yields 0.48% more expected annual energy production for the 11-turbine wind plant than a strategy that neglects uncertainty when considering stochastic inputs. Thus, not only does the present OUU strategy produce more power in realistic conditions, but also it reduces risk by prescribing strategies that call for less extreme yaw angles.

## 1 Introduction

A key determinant in the profitability of a wind power plant is its annual energy production (AEP). The traditional strategy for increasing AEP has been to control each turbine in the plant such that single-turbine power generation is maximized, irrespective of the generation by other turbines. Plant-level control, by contrast, is an innovative approach that has the potential to further optimize wind plant performance and increase AEP (Johnson and Thomas, 2009; Marden et al., 2013; Gebraad et al., 2017; Fleming et al., 2016a; Munters and Meyers, 2018). However, plant-level control presents new challenges in coordinating a set of complex machines, each operating in a highly uncertain and complex flow environment.

Recently, researchers from the National Renewable Energy Laboratory (NREL) have partnered with utility-scale wind power plants to demonstrate the potential benefits of the wind plant control strategy known as wake steering (Flem-ing et al., 2017, 2019). This strategy offsets turbine yaw positions from the incoming wind, thereby "steering" wakes away from downstream turbines (Fleming et al., 2016b; Gebraad et al., 2016; Raach et al., 2016). Accurately characterizing the plant and atmospheric physics is, however, a significant challenge when designing wake steering schemes. In particular, it is difficult to forecast the future behavior of the atmosphere, since the engineering forecast models used in practice are prone to inaccuracies (Nygaard, 2015), infield sensors are subject to bias (Mittelmeier and Kühn, 2018), and many quantities of interest must be extrapolated or interpreted from measured values.

There are also various sources of uncertainty that can have substantial impacts on the success of wake steering strategies. For example, thermally driven vertical mixing in the atmospheric boundary layer is difficult to measure and characterize (Wharton and Lundquist, 2010). Wind speed measurements must also be extrapolated horizontally to forecast conditions far away from sensors, as well as vertically

to characterize the shear in the inflow and wind properties above meteorological measurement tower sensors (Clifton et al., 2016). Moreover, it is common practice to assume a deterministic relationship between turbine power, thrust co-efficients, and wind speed, but there is large scatter in these values when they are measured in practice. Complex phenomena, such as vorticity generated by the turbine blades, cause yaw alignment sensor errors, introducing significant uncertainty in measurements of turbine yaw angles relative to the incoming wind.

Physical uncertainties associated with engineering wake models used for the design of wind plant control strategies pose additional difficulties. For example, although several studies have reported significant gains in AEP using plant-level control strategies under the assumption of perfect (i.e., certain) information (Gebraad et al., 2016; Fleming et al., 2016b; Bossanyi and Jorge, 2016), uncertainties associated with wake model parameters may cause a wake steering strategy in the field to perform differently than anticipated. Gaumond et al. (2014) showed that, by assuming uncertainty in the inflow direction, the predictive capability of engineering wake models may be improved, further emphasizing the importance of uncertainty when developing control strategies.

Uncertainty in the design process can be addressed using optimization under uncertainty (OUU), a technique that has been used in several prior wind plant optimization studies to provide a robust solution under varying levels of uncertainty (Gonzalez et al., 2012; Chen and MacDonald, 2013). Quick et al. (2017) formulated the wake steering problem using OUU, assuming large uncertainties in the yaw positions of individual turbines. Subsequently, Rott et al. (2018) formulated and solved a wake steering OUU problem for a nine-turbine plant, assuming uncertainty in the measured inflow direction. More recently, Simley et al. (2019) formulated an OUU problem by taking yaw position uncertainty and inflow direction variability into account.

In this paper, we extend prior work on OUU and plant-level control to address uncertainty in turbine yaw positions and the direction, speed, shear, and turbulence intensity of the wind inflow during the optimization of turbine yaw off-sets for wake steering strategies. Our objective is to understand how uncertainty in the examined parameters influences wake-steering optimization, and to quantify the effect of uncertainty on the performance of a hypothetical wind power plant. Using a polynomial chaos expansion (PCE) approach, which has not been employed in previous OUU studies of wake steering strategies, we show that direction is the most important uncertain input, effectively smearing out the paths of wakes and reducing the expected velocity deficit. We further show that uncertainty generally reduces optimal yaw off-sets, in agreement with the results of Rott et al. (2018) and Simley et al. (2019) obtained using a simple quadrature approach not based on PCE.

In this study, we first examine a two-turbine test case to explore how different magnitudes of uncertainty impact the efficacy of wake steering schemes, with a particular focus on the trade-off between the power produced by the front and back turbines. Assuming standard uncertainty distributions based on available information, we find that the inflow speed and direction are the most influential parameters to the wake steering design problem. In a more realistic 11-turbine wind plant test case, we further demonstrate the benefits of the OUU formulation. In particular, in addition to yielding more robust designs, the OUU formulation results in less-extreme prescribed yaw offsets than a deterministic problem formulation. Damiani et al. (2018) demonstrated that, without accounting for wakes from upstream turbines, more extreme yaw offsets generally result in more extreme turbine loads.

The paper is organized as follows. In the next section, we outline details of the engineering wake model, the formulation of the OUU problem, and the specific application examined. Results are outlined for two-turbine and wind-plant test cases in Section 3, and conclusions are presented at the end.

## 2   Methodology, application, and approach

In this study, we applied the FLOw Redirection and Induction in Steady-State (FLORIS) engineering wake model (NREL, 2019) to a simple two-turbine test case and to a more realistic 11-turbine wind plant test case to quantify potential benefits of explicitly taking uncertainty into account when designing plant-level wake steering schemes via OUU.

### 2.1   Engineering wake model

We used the FLORIS implementation of the steady-state Gaussian wake model (Bastankhah and Porté-Agel, 2016; Annoni et al., 2018), which imposes a time-independent velocity deficit given by

$$\frac{u(x,y,z)}{u_\infty} = 1 - C \exp\left[-\frac{(y-\delta_c)^2}{2\sigma_y^2} - \frac{(z-z_h)^2}{2\sigma_z^2}\right], \qquad (1)$$

where $u(x,y,z)$ is the velocity component in the direction of the inflow, $x$ is the streamwise direction, $y$ is the cross-flow direction, $z$ is the vertical direction, $\delta_c(x,y,z)$ is the wake deflection field in the crossflow direction, $u_\infty$ is the inflow magnitude at the wind turbine hub height, $z_h$, and $C$ is the velocity deficit in the center of the wake. The standard deviations, $\sigma_y$ and $\sigma_z$, parameterize the width and height of the wake in the crossflow and vertical directions, respectively. Further details on the relationships between the different wake parameters are provided in the documentation for FLORIS (NREL, 2019).

In this study, we limited the value of the thrust coefficient to be strictly less than one. Without this modification, wake calculations for low wind speeds may result in inaccurate predictions (in particular, the calculation of $C$ involves the square root of one minus the thrust coefficient). Throughout this paper, we use the NREL 5-MW reference turbine

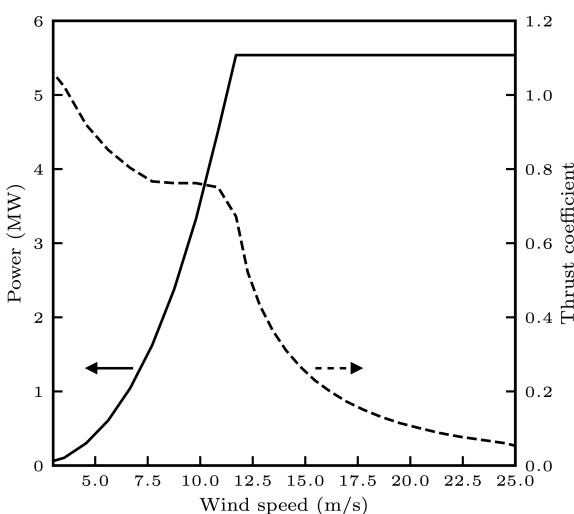

**Figure 1.** Power (solid line) and thrust coefficient (dashed line) as functions of wind speed for the NREL 5-MW reference turbine (Jonkman et al., 2009). Arrows point from the curves to the corresponding vertical axes.

(Jonkman et al., 2009), which has power and thrust coefficient curves shown in Figure 1.

## 2.2 Problem formulation

Using the steady-state FLORIS wake model, the deterministic power production of a wind plant can be predicted given turbine-specific yaw positions, $\boldsymbol{y}$, as well as the average direction, $\theta$, average speed, $u_\infty$, turbulence intensity, TI, and wind shear coefficient, $\alpha$, of the incoming wind over a 10-minute period. We denote the deterministic power prediction from FLORIS as $f(\boldsymbol{v})$, where $\boldsymbol{v} = [\boldsymbol{y}, \theta, u_\infty, \text{TI}, \alpha]$. It should be noted that $\boldsymbol{y}$ is a vector of angular yaw positions for each turbine in a farm and is a relative reference; in this sense, $\boldsymbol{y}$ represents a vector of yaw offsets with respect to $\theta$. The length of the vector, $\boldsymbol{y}$, is equal to the number of turbines in the plant. The inflow direction, $\theta$, is measured clockwise from north and the yaw position is measured counterclockwise from the inflow direction.

During plant operation, low-frequency variation, spatial variability, and measurement errors are inevitable complications that may be represented in a stochastic setting. We envision wake steering strategies changing approximately every 10 minutes. As a result, we introduce the stochastic expected power, denoted $f_{10}$ because it is representative of uncertainties that are relevant on the order of 10 minutes of operational time. It is defined as

$$f_{10} = \int f(\boldsymbol{v}) \, p_v(\boldsymbol{v}) \, d\boldsymbol{v}, \tag{2}$$

where $p_v(\boldsymbol{v})$ is a joint probability density function (pdf) that describes the distribution of $\boldsymbol{v}$, which is representa-

tive of low-frequency temporal variation, spatial variability, and measurement errors. Although this distribution can be empirically determined using real-world measurements and knowledge of turbines in a wind plant, in this study we instead parameterize $p_v$ using the vector of mean values, $\boldsymbol{\mu}_v = [\boldsymbol{\mu}_y, \mu_\theta, \mu_{u_\infty}, \mu_{\text{TI}}, \mu_\alpha]$, where $\mu_a$ denotes an average value of variable $a$, and the hyperparameter vector, $\boldsymbol{\Sigma}$ (which includes, for example, standard deviations if $p_v$ is assumed to be normally distributed). We thus parameterize $f_{10}$ as

$$f_{10}(\boldsymbol{\mu}_v, \boldsymbol{\Sigma}) = \int f(\boldsymbol{v}) \, p_v(\boldsymbol{v}; \boldsymbol{\mu}_v, \boldsymbol{\Sigma}) \, d\boldsymbol{v}, \tag{3}$$

where $p_v(\boldsymbol{v}; \boldsymbol{\mu}_v, \boldsymbol{\Sigma})$ denotes the joint pdf of $\boldsymbol{v}$ parameterized by $\boldsymbol{\mu}_v$ and $\boldsymbol{\Sigma}$. We define this joint pdf such that, as $\boldsymbol{\Sigma} \to \boldsymbol{0}$, $p_v(\boldsymbol{v}; \boldsymbol{\mu}_v, \boldsymbol{\Sigma})$ approaches the Dirac delta function centered on $\boldsymbol{\mu}_v$, namely $\delta(\boldsymbol{v} - \boldsymbol{\mu}_v)$.

The energy production may be estimated for a whole year (i.e., the expected AEP) as a linear sum of each speed- and direction-specific expected power production, weighted by speed- and direction-specific probabilities and multiplied by 8,760 hours per year. These probabilities are representative of annual variability, as opposed to the previously described uncertainty in operating conditions. Thus, the average inflow speed and direction are cast as being uncertain in order to capture their annual variability. In practice, these probabilities are empirically determined and jointly distributed. The resulting expression for AEP is thus given as

$$\text{AEP}(\boldsymbol{\mu}_y, \mu_{\text{TI}}, \mu_\alpha, \boldsymbol{\Sigma}) =$$

$$8760 \int f_{10}(\boldsymbol{\mu}_v, \boldsymbol{\Sigma}) \, p_\mu(\mu_{u_\infty}, \mu_\theta) d\mu_{u_\infty} d\mu_\theta, \tag{4}$$

where $p_\mu(\mu_{u_\infty}, \mu_\theta)$ represents the joint distribution of $\mu_{u_\infty}$ and $\mu_\theta$ over a year.

Using Eq. (4) for the AEP, we can formulate the wake steering OUU problem as

$$\boldsymbol{\mu}_y^{(\text{OUU})} = \arg\max_{\boldsymbol{\mu}_y} \text{AEP}(\boldsymbol{\mu}_y, \mu_{\text{TI}}, \mu_\alpha, \boldsymbol{\Sigma}). \tag{5}$$

Similarly, the deterministic wake steering optimization is formulated for $\boldsymbol{\Sigma} = \boldsymbol{0}$ as

$$\boldsymbol{\mu}_y^{(\text{det})} = \arg\max_{\boldsymbol{\mu}_y} \text{AEP}(\boldsymbol{\mu}_y, \mu_{\text{TI}}, \mu_\alpha, \boldsymbol{0}). \tag{6}$$

The baseline solution corresponds to turbines that are directly aligned with $\theta$ such that there is no yaw offset, corresponding to

$$\boldsymbol{\mu}_y^{(\text{base})} = \boldsymbol{0}. \tag{7}$$

We used four metrics to assess the quality of different solutions for $\boldsymbol{\mu}_y$. The value of the stochastic solution (VSS) is the expected value of the stochastic AEP for the OUU solution relative to the deterministic solution. Our VSS definition is similar to the VSS metric introduced by Birge and

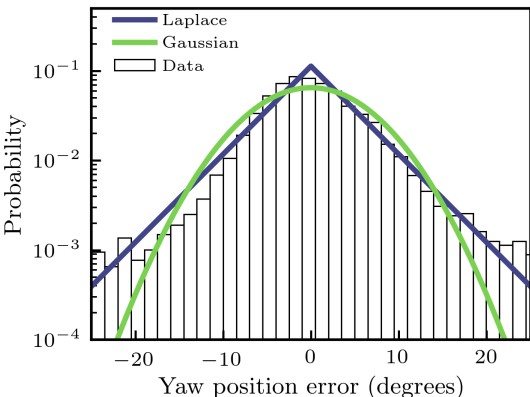

**Figure 2.** Errors in the yaw position, $\boldsymbol{y}$, for a test turbine at the National Wind Technology Center at NREL (Fleming et al., 2018; Annoni et al., 2018; Damiani et al., 2018). Measurement data used to calculate the probability distribution were sampled every 30 s. The solid blue and green lines show Laplace and Gaussian distributions, respectively. The empirical probability mass function found from the observed yaw errors is shown with white bars.

Louveaux (2011), but is expressed as a fractional increase in expected AEP rather than an absolute value increase. As a result, the solution value metrics do not depend on the amount of power produced. We also examined the expected value of stochastic AEP for the OUU solution relative to the baseline no-offset case, denoted $\text{VSS}_b$. The value of the deterministic solution (VDS) is the nonstochastic value of the AEP for the deterministic solution relative to the baseline solution. In addition, we report the stochastic value of the AEP for the deterministic solution relative to the baseline solution, denoted as $\text{VDS}_s$. Each of these metrics is defined in Table 1.

### 2.3 Application

#### 2.3.1 Uncertainty estimates

In the present demonstration tests, we considered the effects of uncertainty in turbine yaw offsets and wind inflow speed, direction, turbulence intensity, and shear. We envision wake steering strategies changing every 10 or 20 minutes, so we identify reasonable uncertainty values to represent spatial and temporal variations, as well as measurement errors, in each of these uncertain parameters over that time span. Together, these variations comprise the joint pdf, $p_v(\boldsymbol{v}; \boldsymbol{\mu}_v, \boldsymbol{\Sigma})$. It should be noted that the present formulation and demonstration of the analysis approach is not specific to the uncertainty values used here, and the method is equally valid for other choices of these values that may represent different real-world conditions and wind plants.

To estimate the yaw position uncertainty, we compared operational data from an NREL turbine with a nearby meteorological measuring mast (NWTC Information Portal, 2019); these data were examined previously by Fleming et al.

(2018), Annoni et al. (2018), and Damiani et al. (2018). In the present study, the wind direction recorded at the turbine was compared to the wind direction measured on the upstream meteorological mast. The mean error, which is sometimes referred to as bias, was removed to focus on the shape of the distribution of errors, as shown in Figure 2.

Based on the shape of the distribution in Figure 2, we parameterize the yaw misalignment as a two-sided exponential distribution, termed the Laplace distribution, given by

$$L(x; \mu, \nu) = \frac{1}{2\nu} \exp\left(-\frac{|x - \mu|}{\nu}\right), \tag{8}$$

where $\mu$ is the mean and $\nu$ is a scale parameter. The remaining uncertain parameters are assumed to be normally distributed according to

$$N(x; \mu, \sigma) = \frac{1}{\sqrt{2\pi\sigma^2}} \exp\left[-\frac{(x - \mu)^2}{2\sigma^2}\right], \tag{9}$$

where $\mu$ is again the mean and $\sigma$ is the standard deviation. Each of the uncertain parameters is then assumed to be independent such that the joint pdf, $p_v(\boldsymbol{v}; \boldsymbol{\mu}_v, \boldsymbol{\Sigma})$, can be written as

$$\begin{aligned} p_v(\boldsymbol{v}; \boldsymbol{\mu}_v, \boldsymbol{\Sigma}) = \\ [L(\boldsymbol{y}; \boldsymbol{\mu}_y, \boldsymbol{\nu}_y) N(\theta; \mu_\theta, \sigma_\theta) N(u_\infty; \mu_{u_\infty}, \sigma_{u_\infty}) \\ N(\text{TI}; \mu_{\text{TI}}, \sigma_{\text{TI}}) N(\alpha; \mu_\alpha, \sigma_\alpha)], \end{aligned} \tag{10}$$

where the hyperparameter is given as $\boldsymbol{\Sigma} = [\boldsymbol{\nu}_y, \sigma_\theta, \sigma_{u_\infty}, \sigma_{\text{TI}}, \sigma_\alpha]$. The vector $\boldsymbol{\nu}_y$ represents the scale parameter used in the yaw offset Laplace distributions for each turbine in a plant. It should be noted that Mittelmeier and Kühn (2018) reported yaw misalignment to be a strong function of the inflow wind speed, which is not considered here.

Estimated values for $\boldsymbol{\Sigma}$ are taken from a range of sources. Based on the observational data shown in Figure 2, we measured a scale parameter of $\nu = 6.16°$, and we correspondingly set $\boldsymbol{\nu}_y = 5°$ for all turbines. Mittelmeier et al. (2017) discuss a methodology to estimate inflow conditions from turbine sensor data. They reported Gaussian uncertainties of $3.6°$ and $0.46$ m/s when predicting the inflow direction and speed, respectively. Similarly, Gaumond et al. (2014) provided direction variations measured over a 10-minute interval in the Horns Rev power plant, which yielded a standard deviation of $2.67°$. Based on these studies, we propose $\sigma_\theta = 5°$ and $\sigma_{u_\infty} = 1$ m/s as reference uncertainty values. Uncertainties for the shear parameter, $\alpha$, and the turbulence intensity, TI, are more difficult to determine and have received relatively little attention in previous studies. Consequently, for the present demonstration of the OUU method, we have selected relatively large uncertainties of $\sigma_\alpha = 0.05$, which is just under half as large as the assumed mean value of $\mu_\alpha = 0.12$ used in the present tests, and $\sigma_{\text{TI}} = 5\%$, which is close to the mean value of $\mu_{\text{TI}} = 6\%$ used in these tests.

**Table 1.** Summary of AEP-based metrics used to assess the quality of solutions for given values of $\mu_{\text{TI}}$ and $\mu_\alpha$ (for simplicity, these parameters are suppressed as arguments of the AEP in the notation).

| Metric | Equation | Description |
|---|---|---|
| VSS | $\text{AEP}[\boldsymbol{\mu}_y^{(\text{OUU})}, \boldsymbol{\Sigma}]/\text{AEP}[\boldsymbol{\mu}_y^{(\text{det})}, \boldsymbol{\Sigma}] - 1$ | Stochastic AEP from OUU relative to the deterministic solution |
| $\text{VSS}_b$ | $\text{AEP}[\boldsymbol{\mu}_y^{(\text{OUU})}, \boldsymbol{\Sigma}]/\text{AEP}[\mathbf{0}, \boldsymbol{\Sigma}] - 1$ | Stochastic AEP from OUU relative to the baseline solution |
| VDS | $\text{AEP}[\boldsymbol{\mu}_y^{(\text{det})}, \mathbf{0}]/\text{AEP}[\mathbf{0}, \mathbf{0}] - 1$ | Nonstochastic AEP from the deterministic solution relative to the baseline solution |
| $\text{VDS}_s$ | $\text{AEP}[\boldsymbol{\mu}_y^{(\text{det})}, \boldsymbol{\Sigma}]/\text{AEP}[\mathbf{0}, \boldsymbol{\Sigma}] - 1$ | Stochastic AEP from the deterministic solution relative to the baseline solution |

**Table 2.** Probability distributions and hyperparameter values describing the uncertainty associated with various inputs to the wake model.

| Parameter | Distribution | Hyperparameter |
|---|---|---|
| Yaw offsets, $\boldsymbol{y}$ | $L(\boldsymbol{y}; \boldsymbol{\mu}_y, \boldsymbol{\nu}_y)$ | $\boldsymbol{\nu}_y = 5°$ |
| Wind direction, $\theta$ | $N(\theta; \mu_\theta, \sigma_\theta)$ | $\sigma_\theta = 5°$ |
| Wind speed, $u_\infty$ | $N(u_\infty; \mu_{u_\infty}, \sigma_{u_\infty})$ | $\sigma_{u_\infty} = 1$ m/s |
| Turbulence, TI | $N(\text{TI}; \mu_{\text{TI}}, \sigma_{\text{TI}})$ | $\sigma_{\text{TI}} = 5\%$ |
| Shear, $\alpha$ | $N(\alpha; \mu_\alpha, \sigma_\alpha)$ | $\sigma_\alpha = 0.05$ |

Although these uncertainties are likely unrealistically large, it will be seen in the two-turbine test case that even these large uncertainties have less of an effect on the wake steering problem than uncertainties in $\boldsymbol{y}$, $\theta$, and $u_\infty$.

The resultant distribution choices and hyperparameter estimates are provided for each uncertain variable in Table 2. It is cautioned that the magnitude of these sources of uncertainty are site-specific. For example, a wind plant built in the wake of a large obstacle would be expected to have larger uncertainty in the inflow direction than a wind plant built offshore. As such, the uncertainties outlined in Table 2 should be taken as representative of real uncertainties but do not correspond to any particular site or wind plant. In the real world, the uncertainties are likely to differ for each wind power plant, since there are many different potential sources of uncertainty in the parameters necessary to initialize and solve the FLORIS model for a wind plant. Correlations between different parameters and their uncertainties, such as between wind speed and shear, turbulence intensity, or yaw error, will also affect the results, but are not considered here.

### 2.3.2 Calculation of AEP

We approximated the integral in Eq. (3) for $f_{10}$ using PCE, which uses orthogonal polynomials with collocated quadrature points to interpolate a quantity of interest through an uncertain parameter space (Eldred and Elman, 2011). We used the PCE tool in DAKOTA (Adams et al., 2014) in all cases. As DAKOTA does not support Laplace distributions, we report the mean response resulting from all combinations of exponentially distributed yaw position errors.

PCE was used to facilitate computing $f_{10}$ instead of relying on simple quadrature, which requires a very fine-spaced grid of inputs, or Monte Carlo simulation, which requires

on the order of millions of simulation evaluations. Our approach integrates the PCE surrogate when estimating $f_{10}$, effectively replacing low-order interpolating functions used in simple quadrature with a surrogate model that requires fewer quadrature points in order to achieve the same level of accuracy. Padrón et al. (2019) recently demonstrated the advantages of PCE in computing AEP as opposed to the traditional simple quadrature. When computing the integral in the two-turbine cases, we used fifth-order quadrature with uniform $p$-refinement and two maximum refinement iterations. In the OUU problem, we used fifth-order quadrature for the variable products without refinement during each optimization iteration and used fifth-order quadrature with $p$-refinement and two maximum refinement levels to assess the outcome of the wind plant OUU, deterministic, and baseline optimization solutions. For more details regarding PCE, we refer the reader to the DAKOTA theory manual (Adams et al., 2014).

During computation of the AEP via Eq. (4), the speed and direction joint pdf, $p_\mu(\mu_\theta, \mu_{u_\infty})$, is approximated with an empirical discrete joint probability mass function, denoted $\rho_\mu(\mu_\theta^d, \mu_{u_\infty}^i)$. Here, $d = [1, \ldots, D]$ and $i = [1, \ldots, I]$, where $D$ is the number of directional bins and $I$ is the number of inflow wind speed bins in the discrete function $\rho_\mu$. This discretization thus yields a new definition of AEP, given as

$$\text{AEP}(\boldsymbol{\mu}_y, \mu_{\text{TI}}, \mu_\alpha, \boldsymbol{\Sigma}) =$$
$$8760 \sum_{d=1}^{D} \sum_{i=1}^{I} \rho(\mu_\theta^d, \mu_{u_\infty}^i) f_{10}\left(\left[\boldsymbol{\mu}_y, \mu_\theta^d, \mu_{u_\infty}^i, \mu_{\text{TI}}, \mu_\alpha\right], \boldsymbol{\Sigma}\right).$$

(11)

### 2.3.3 Layouts considered

To demonstrate the benefits of OUU in the development of wake steering strategies, we considered a two-turbine layout as well as a larger 11-turbine layout. We used the two-turbine layout to explore the basic trade-off between the power production of front and back turbines as well as the sensitivity to different levels of uncertainty. The wind plant problem was used to assess the potential benefits of OUU in a more realistic wind plant design problem. The mean values, scale parameters, and upper and lower bounds associated with each input considered in the two-turbine and wind plant cases are shown in Table 3.

In the two-turbine case, the front turbine directly wakes the back turbine when flow is from the north, as shown in Fig-

**Table 3.** Mean and scale values and lower and upper bounds used in the two-turbine and wind plant OUU problems. Sequences are expressed as [start : increment : end].

| Parameter $i$ | Two-turbines $\mu_i$ | Two-turbines $\Sigma_i$ | Plant $\mu_i$ | Plant $\Sigma_i$ | Lower Bound | Upper Bound |
|---|---|---|---|---|---|---|
| $\boldsymbol{y}$ (°) | [-30:1:30] | [1:1:15] | - | 0 | $-\infty$ | $\infty$ |
| $\theta$ (°) | [-60:10:60] | [1:1:15] | [-60:10:60] | 5 | $-\infty$ | $\infty$ |
| $u_\infty$ (m/s) | [3:1:15] | [0.2, 0.5, 1, 2] | [3:1:15] | 1 | 3 | 20 |
| TI (%) | 6 | [1:1:10] | 6 | 0 | 1 | 30 |
| $\alpha$ | 0.12 | [0.02, 0.05, 0.1, 0.15, 0.2] | 0.12 | 0 | -0.5 | 3.0 |

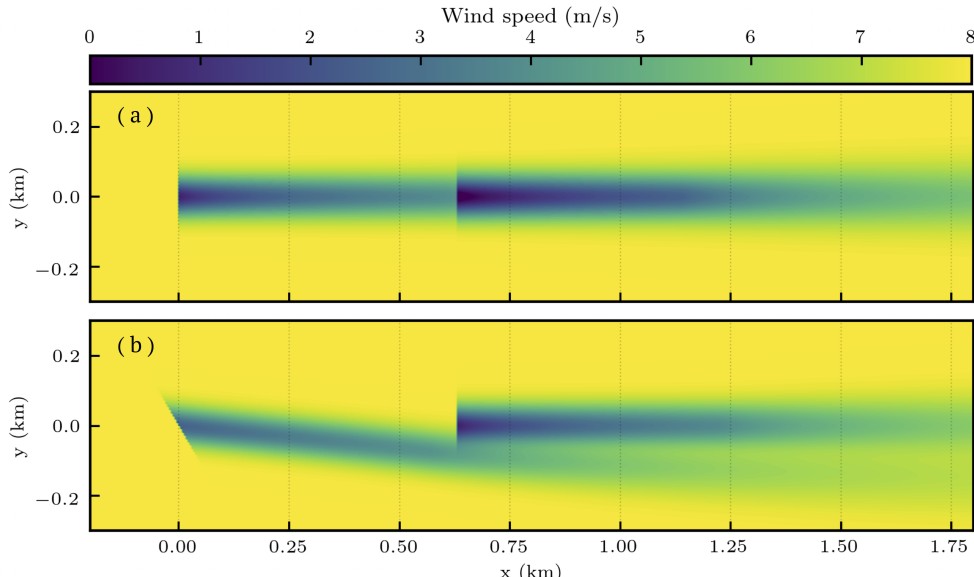

**Figure 3.** Contours of wind speed for a simple two-turbine test case with an inflow speed of 8 m/s. Brighter colors correspond to faster wind speeds. In (a), the turbines are both directly facing the wind with $\boldsymbol{y}$ set to $\boldsymbol{0}$. In (b), the front turbine is offset such that $y_1 = 30°$ and $y_2 = 0°$.

ure 3(a). The turbines are spaced five rotor diameters apart in the northern direction. We chose this case because it is representative of the fundamental trade-off between upstream turbines losing power by offsetting their yaw positions and 5 downstream turbines gaining power when wakes are diverted away from them [as indicated in Figure 3(b)]. We performed a parameter sweep across possible values of the front turbine yaw offset with a nested PCE routine to find the optimum steering strategy for various uncertainties in the inflow. We 10 report the maximum VSS across all directions and speeds for each uncertain input using the reference scale values. Uncertain variables associated with larger maximum VSS are assumed to be more important and, based on the results from this two-turbine case, we included only the most important 15 uncertainties in the more computationally expensive wind plant case.

The wind plant wake steering optimization problem is intended to provide insights on the benefits of OUU in more realistic scenarios. The plant layout is shown in Figure 4, and 20 the corresponding annual wind speed and direction probability mass function is shown in Figure 5. We performed deter-

ministic and stochastic wake steering optimizations for each speed and direction, reporting the deterministic and expected power production associated with the OUU, deterministic, and baseline strategies. We used the annual wind speed and 25 direction probability mass function to aggregate these speed- and direction-specific power production estimates into an estimate of AEP.

It should be noted that a relatively limited range of inflow directions and relatively large direction increments were con- 30 sidered in the present study, as shown in Figure 5. These choices were made to simplify the demonstration case, and the approach is readily extended to wider inflow direction ranges with smaller increments. In particular, optimization of AEP with respect to both wind turbine layout and yaw po- 35 sitions would require the use of smaller inflow increments. Data for the joint distribution of the wind speed and direction shown in Figure 5 are provided in the supplemental material.

The expected power production was maximized during the optimization. The COBYLA optimization driver in 40 DAKOTA (Adams et al., 2014) was used to design the wake steering strategies. The PCE tool in DAKOTA (Adams et al.,

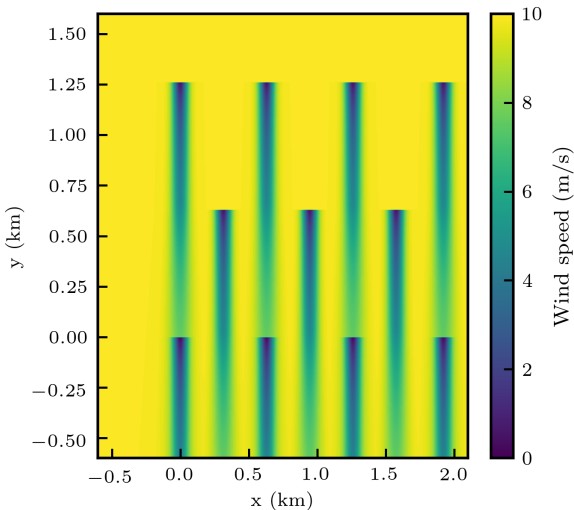

**Figure 4.** Contours of wind speed for the 11-turbine wind farm with the baseline yaw configuration ($\boldsymbol{y} = \boldsymbol{0}$) and 10 m/s inflow speed. Brighter colors correspond to faster wind speeds.

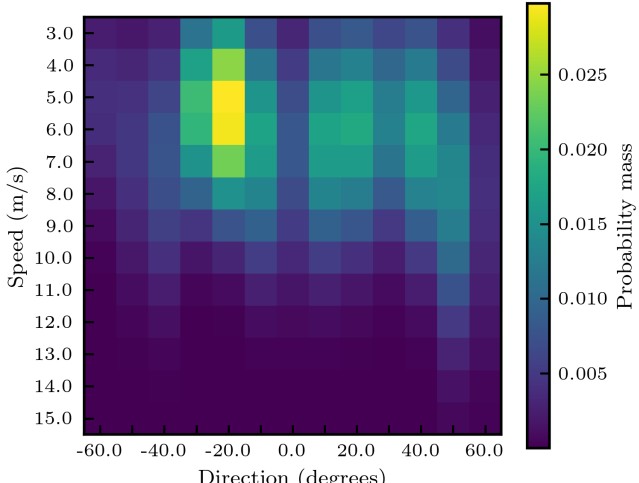

**Figure 5.** Annual wind speed and direction probability mass function used in the 11-turbine wind plant optimization study.

2014) was used during each optimization iteration to estimate the stochastic response in the OUU. Each OUU was initialized with the corresponding deterministic solution.

## 3  Results

In the following, we present results for OUU of the simple two-turbine case as well as the 11-turbine wind plant. It will be shown from an analysis of the two-turbine case that wind speed and direction are the most influential parameters, and so we performed the plant OUU using only these two uncertain variables, assuming $\boldsymbol{\nu}_y$, $\sigma_{\mathrm{TI}}$, and $\sigma_\alpha$ to be zero. Optimization results of wake steering strategies for the 11-turbine

**Table 4.** Maximum VSS across all speeds and directions considered, given the reference standard deviation values in Table 2.

| Parameter | $\boldsymbol{y}$ | $\theta$ | $u_\infty$ | TI | $\alpha$ |
|---|---|---|---|---|---|
| $\max(\mathrm{VSS}\vert\boldsymbol{\Sigma})$ | 0.32% | 5.4% | 0.60% | 0.28% | 0.02% |

wind plant are presented using the OUU and deterministic problem formulations, and the results are compared to baseline strategies (i.e., using no wake steering).

### 3.1  Two-turbine test case

Figure 3 shows results for the two-turbine test case, where the front turbine wakes the back turbine. For each uncertain parameter, we performed a parameter sweep across possible values of the front turbine yaw offset with a nested sampling routine to find the optimum steering strategy for various levels of uncertainty. The results are summarized in Table 4. Based on these results, uncertainty in the wind direction, $\theta$, was found to give the maximum VSS value, followed by uncertainties in the wind speed, $u_\infty$, yaw position, $\boldsymbol{y}$, and turbulence intensity, TI. Uncertainty in shear, $\alpha$, was found to have very little impact for wake steering designs.

Uncertainty in the wind direction affects the path that wakes behind wind turbines will follow. This can be thought of as spreading out the wake. This effect is explored in Figure 6, which shows that, as the inflow direction uncertainty increases, the wake becomes spread out such that the power of the back turbine is eventually completely insensitive to the yaw angle of the front turbine. The effect of uncertainty in direction on the front turbine optimal yaw settings is dramatic. For example, in Figure 6(c), the optimal yaw offset is around 25° when there is perfect information. As mild uncertainty is introduced, however, the optimal front turbine yaw angle decreases. When large levels of uncertainty are introduced, the optimal setting switches to almost no steering. The optimal front turbine yaw offset is shown as a function of inflow direction for different levels of uncertainty in Figure 6(d). Once again, as uncertainty increases, the optimal yaw offset becomes more gradual and less extreme.

It is interesting to note that the deterministic solution may be worse than the baseline solution if there is large uncertainty in the inflow wind direction. This is shown in Figure 7, which indicates that, as inflow direction uncertainty increases, there is less overall benefit to wake steering. Results for $\mathrm{VSS}_b$ in Figure 7(b) show that the increase in power production is reduced from around 10% to 1% as $\sigma_\theta$ increases from 1° to 15°. The VDS results in Figure 7(c) have a maximum of almost 15% and, by definition, are not affected by uncertainty. We found that the deterministic strategy performed on the order of 10% worse than the baseline solution for large levels of direction uncertainty, which may be observed in the $\mathrm{VDS}_s$ results shown in Figure 7(d).

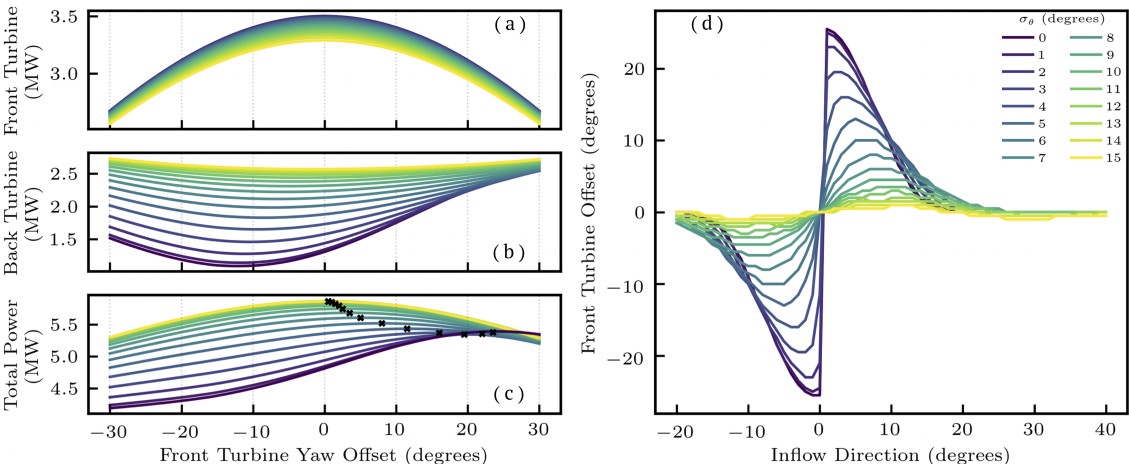

**Figure 6.** Power production for the front (a), back (b), and both (c) turbines as a function of front turbine yaw offset in the two-turbine case with 10 m/s inflow $3^\circ$ from north. Different line colors indicate different values of $\sigma_\theta$, as indicated in panel (d), with brighter colors corresponding to larger $\sigma_\theta$ and, hence, greater uncertainty in inflow direction. The black crosses in panel (c) denote optimal front turbine yaw settings. Panel (d) shows the optimal front turbine yaw offset as a function of inflow direction for 7 m/s inflow.

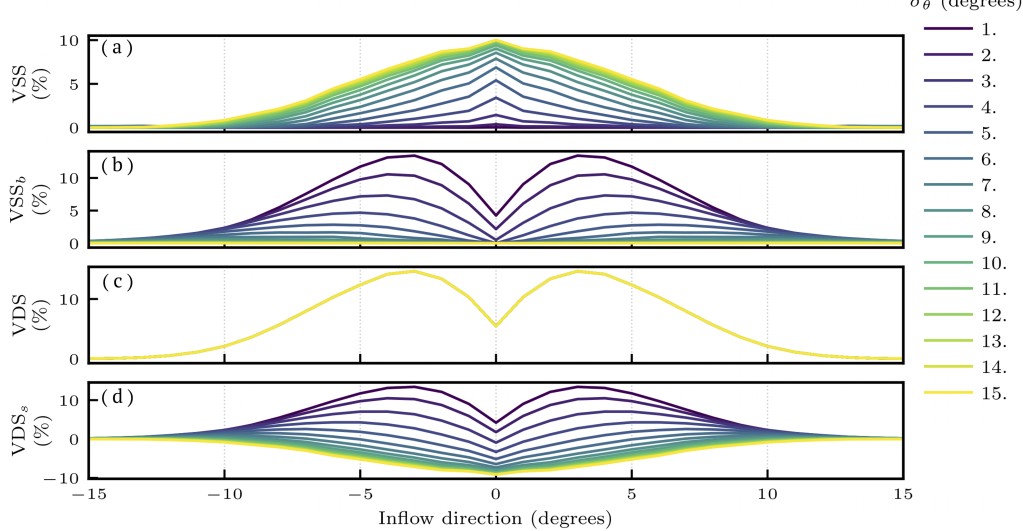

**Figure 7.** Summary statistics from Table 1 for varying inflow direction $\theta$ and uncertainty $\sigma_\theta$ in the two-turbine case with 7 m/s inflow. Panels show VSS (a), $\mathrm{VSS}_b$ (b), VDS (c), and $\mathrm{VDS}_s$ (d), with different values of $\sigma_\theta$ indicated by different line colors (the legend is shown at the right of the figure).

Uncertainty in the incoming wind speed, $u_\infty$, changes the magnitude of the wake velocity deficits, although the wake paths remain unchanged. Variability in the wind speed leads to power fluctuations around the nominal power curve, with different behaviors depending on the operating condition. Wind speed variability in the region where power production tracks the cubic power of velocity generally leads to higher than nominal power production. By contrast, wind speed variability in the region where the power is limited by the generator size generally leads to lower than nomi-

nal power production. This is due to Jensen's inequality and the concavity of different sections of the power curve (Quick et al., 2016). In either case, the difference in expected power production will result in different deterministic and stochastic operational strategies. Figure 8 shows that uncertainty in lower inflow speeds caused the optimal front turbine offset to decrease, and uncertainty in higher wind speeds caused the optimal front turbine offset to increase. The optimal front turbine offset was insensitive to wind speed uncertainty in the cubic range of the power curve. The increased uncertainty in

the inflow speed changes expected power production, which changes the trade-off between reduced power from the upstream turbine and increased power from the downstream turbine due to yaw deflection.

Overall, we found this two-turbine case to be less sensitive to uncertainties in yaw misalignment, turbulence intensity, and wind shear. Uncertainty in the turbine yaw positions generally reduces the rotor-swept area and spreads out the path of the turbine wake. As a result, the power of the back turbine may be increased or decreased by yaw misalignment uncertainty, depending on which dynamic dominates. Yaw position uncertainty does not dramatically affect the solution at the reference uncertainty ($\nu_y = 5°$), but produces noticeably different solutions near $\nu_y = 10°$, which has a maximum VSS of 1.23%.

Turbulence intensity affects the wake expansion geometry, which effectively smears out the path of wakes, decreasing the velocity deficit felt by waked turbines. Although we did not find turbulence intensity uncertainty to be significant here, we found a maximum VSS of 1.29% when $\sigma_{TI} = 10\%$. Introducing Gaussian uncertainty in the shear coefficient did not affect the optimum front turbine angle beyond one or two degrees, even at dramatic levels of uncertainty.

## 3.2   Wind plant test case

To quantify the benefits of OUU in a more realistic scenario, we also performed OUU to design wake steering strategies for an 11-turbine wind power plant. Given the maximum VSS results summarized in Table 4 for the two-turbine case, we considered uncertainties only in the wind inflow direction and speed for the 11-turbine wind plant case. This decision was driven by both the relative impacts of different uncertainties, as well as by the computational cost associated with accounting for each uncertainty. In particular, the maximum VSS for $y$ is roughly half that for $u_\infty$, which is itself nearly an order of magnitude smaller than the maximum VSS for $\theta$. However, consideration of yaw offsets for each turbine in the wind plant case increases the stochastic problem dimension by 11, resulting in substantial additional computational expense in the OUU approach. Consequently, we only considered uncertainty in $u_\infty$ and $\theta$ for the wind plant case, with the understanding that the approach is readily extended to include other sources of uncertainty, given sufficient computational resources.

The stochastic average and deterministic AEP associated with the OUU, deterministic optimization, and baseline (i.e., no wake steering) solutions are provided in Table 5. These represent the aggregate of the different optimization solutions, where powers are weighted by the speed- and direction-specific annual probabilities of occurrence shown in Figure 5. Table 5 shows that, given perfect information, the deterministic strategy is expected to produce 2.6% more AEP than the baseline strategy. However, for the present assumed input uncertainties, the deterministic strategy may be

expected to perform comparably to the baseline strategy, and the OUU strategy may be expected to produce 0.58% and 0.48% more AEP than the baseline and deterministic strategies, respectively.

It is interesting to note that the uncertain expected AEP is greater than the deterministic AEP in Table 5 for all three strategies. This represents the aggregate across the annual wind speed and direction probability mass function. We found that lower wind speeds (below 12 or 13 m/s) were generally associated with increased power production from uncertainty, while larger wind speeds yielded an expected power less than the deterministic value. When we only considered direction uncertainty, the expected power was consistently larger than its deterministic counterpart. This is because the wakes are inherently spread out by uncertainty in direction, reducing the expected velocity deficit in waked regions. When we only consider wind speed uncertainty, larger wind speeds were associated with decreased expected power, and smaller wind speeds were associated with increased expected power. This matches the intuition from Jensen's inequality discussed earlier.

Figure 9 summarizes improvements in AEP for the different wake steering strategies for varying wind speed and direction. Some strategies appear to produce more than 15% more power, given perfect inflow information [reflected in the VDS results in Figure 9(c)], but these same strategies produce almost 2% less power than the baseline no-steering strategy under uncertain conditions [shown in the VDS$_s$ results in Figure 9(d)]. The VSS$_b$ and VDS$_s$ metrics in Figures 9(b) and (d), show that some deterministic and OUU solutions may produce 2% and 4% improvements in average power production, respectively, which is much lower than the increase predicted by the deterministic scenarios indicated in the VDS results shown in Figure 9(c). The optimization histories of the OUU and deterministic approaches are shown for 12 m/s inflow 30° from north in Figure 10.

In general, we found that by incorporating uncertainty in the wake steering problem formulation, less extreme yaw offsets were required to optimize AEP. We show the aggregate of yaw positions suggested by the OUU and deterministic optimization approaches in Figure 11. Although the histogram in Figure 11 is not weighted by probability of inflow occurrence, these results nevertheless strongly suggest that wind plant designers may expect OUU to yield wake steering strategies with lower-magnitude yaw offsets than when using the deterministic optimization formulation.

## 4   Conclusions

In this study, we examined how uncertainty affects wake steering strategies and what benefits may be associated with designing these strategies in the presence of operational uncertainty using OUU. While previous approaches (Simley et al., 2019; Rott et al., 2018) have used simple quadrature

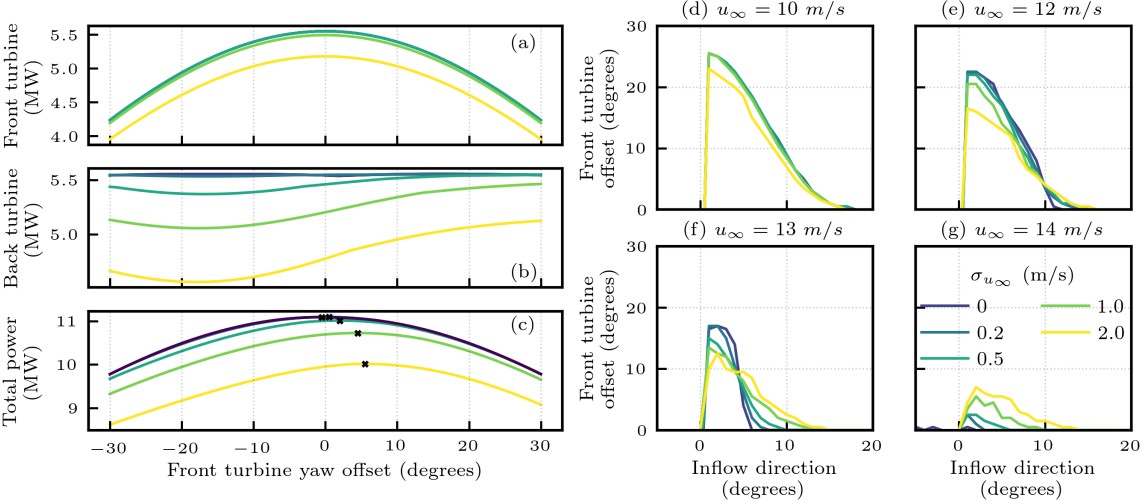

**Figure 8.** Power production for the front (a), back (b), and both (c) turbines as a function of front turbine yaw offset in the two-turbine case with 13 m/s inflow 7° from north. Different line colors indicate different values of $\sigma_{u_\infty}$, as indicated by the legend in panel (g), with brighter colors corresponding to larger $\sigma_{u_\infty}$ and, hence, greater uncertainty in inflow speed, $u_\infty$. The black crosses in panel (c) denote optimal front turbine yaw settings. Panels (d–g) show the optimal front turbine yaw offset as a function of inflow direction for inflow speeds, $u_\infty$, of 10, 12, 13, and 14 m/s, respectively. Only positive inflow directions are shown to highlight the important effects of uncertainty.

**Table 5.** Expected and deterministic AEP of OUU, deterministic, and baseline plant-level wake steering strategies for the 11-turbine wind plant test case.

|                            | Expected AEP (GWh) | Deterministic AEP (GWh) |
|----------------------------|--------------------|-------------------------|
| OUU optimization           | 115.2              | 113.3                   |
| Deterministic optimization | 114.7              | 114.0                   |
| Baseline strategy          | 114.6              | 111.1                   |

to estimate AEP, we demonstrated the more efficient PCE approach. We show that uncertainty in inflow direction effectively smears out the wake. Although the yaw positions are not explicitly considered in the objective function, the OUU formulation leads to smaller yaw offsets, which should generally lead to less extreme loads.

Uncertainty in yaw positions is epistemic and may be reduced with more accurate yaw position detection methods. Uncertainty in inflow conditions is more nuanced. While there are issues with accurately measuring these quantities, fundamentally, there may not be a single characteristic direction, speed, turbulence intensity, or shear associated with the wind flowing into a utility-scale wind plant. For example, a wind power plant may be built downstream of a mountain, causing wind to enter from multiple directions. So, the uncertainties in these inflow parameters may be thought of as a combination of epistemic and aleatoric, irreducible, or model-form uncertainties.

The fact that OUU results in more expected power production with less extreme yaw offsets makes a strong case for designers to move toward OUU formulations in plant-level control strategies. In particular, OUU results in wake steering strategies that are more conservative than the determin-

istic approach — the magnitude of the turbine yaw offsets determined by OUU is diminished, compared to those found using deterministic optimization. Assuming that the inflow uncertainties were precisely quantified, we have shown that wake strategies formulated with the OUU approach can produce up to roughly 4% more power (for the 5-m/s wind speed and 0° wind direction) than wake steering strategies formulated using the deterministic approach.

We are optimistic for the future of plant control strategies and anticipate that uncertainty will become increasingly incorporated in future plant control analysis. In future work, we plan to further quantify typical levels of uncertainty in input parameters, explore higher-fidelity flow models, and to include fatigue loading in the OUU objective function. There are several other sources of uncertainty that may be injected into this problem. For example, we assumed perfect knowledge of the turbine power and thrust curves. Typical levels of uncertainty in turbine power and thrust curves probably would have resulted in somewhat different optimum wake steering strategies. Uncertainty in power would change the shape of the mean power curve, altering the trade-off point, and uncertainty in thrust would affect the magnitude of the velocity deficit behind the turbine. Quantifying fatigue load-

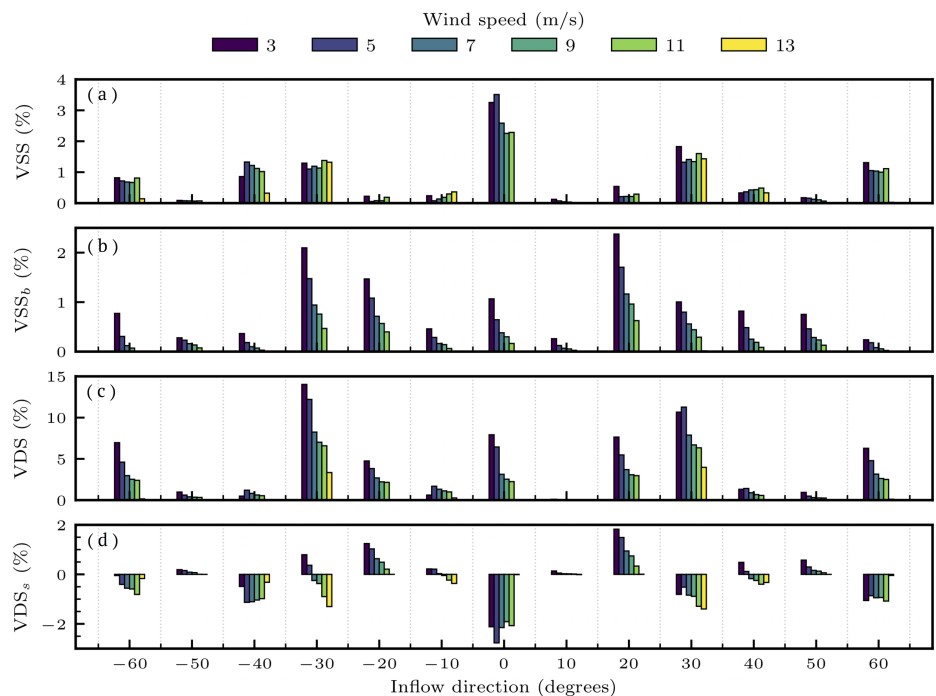

**Figure 9.** Summary statistics from Table 1 for varying inflow direction, $\theta$, and wind speed, $u_\infty$, in the 11-turbine wind plant test case. Panels show VSS (a), $\text{VSS}_b$ (b), VDS (c), and $\text{VDS}_s$ (d), with different values of $u_\infty$ indicated by different line colors (the legend is shown at the top).

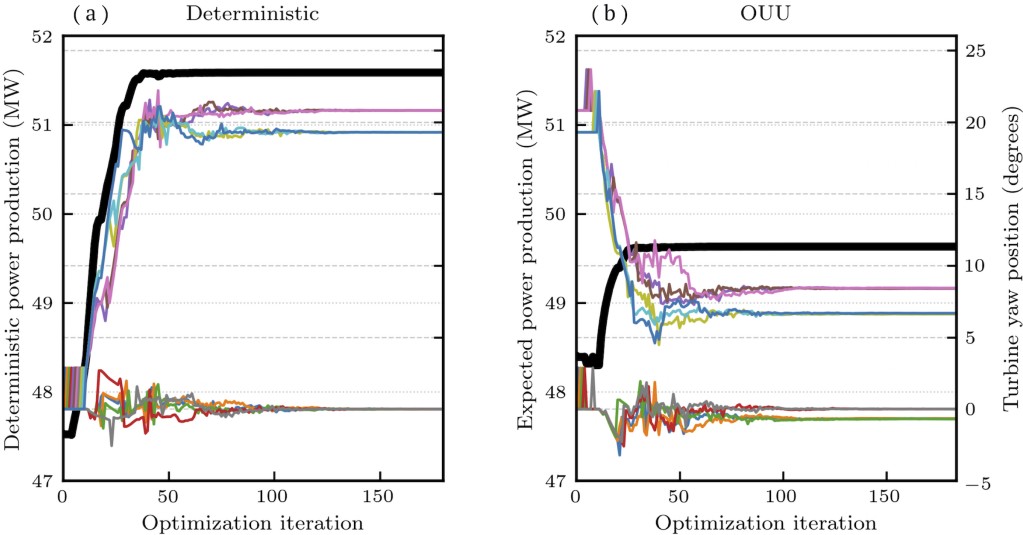

**Figure 10.** Optimization histories for deterministic (a) and stochastic (b) 11-turbine wind plant optimization studies with 12 m/s inflow 30° from north. The deterministic optimization was initialized with the baseline strategy, and the stochastic optimization was initialized with the deterministic solution. The thick black line in each panel shows power (left vertical axes), and the thin multicolored lines show the yaw positions of the different turbines (right vertical axes).

ing is an attractive prospect, though it requires a more advanced wake model. Partial waking may be more detrimental than full exposure to a wake, complicating the fundamental trade-offs that we explored. Finally, the effect of correlations between different sources of uncertainty also deserves further future consideration.

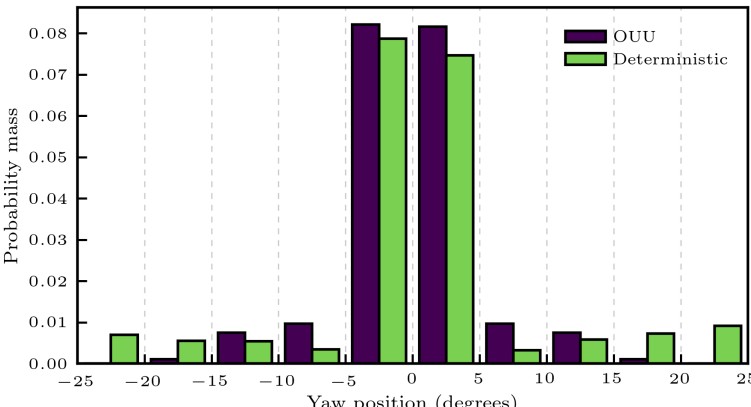

**Figure 11.** Probability mass distributions of $y$ summed over all turbines, wind speeds, and wind directions in the 11-turbine wind plant test case, as prescribed by the OUU and deterministic wind plant optimization strategies.

**Acknowledgements.** This work was authored in part by the National Renewable Energy Laboratory, operated by Alliance for Sustainable Energy, LLC, for the U.S. Department of Energy (DOE) under Contract No. DE-AC36-08GO28308. Funding provided by the U.S. Department of Energy Office of Energy Efficiency and Renewable Energy Wind Energy Technologies Office. The views expressed in the article do not necessarily represent the views of the DOE or the U.S. Government. The U.S. Government retains and the publisher, by accepting the article for publication, acknowledges that the U.S. Government retains a nonexclusive, paid-up, irrevocable, worldwide license to publish or reproduce the published form of this work, or allow others to do so, for U.S. Government purposes. The research was performed using computational resources sponsored by the U.S. Department of Energy's Office of Energy Efficiency and Renewable Energy and located at the National Renewable Energy Laboratory.

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
