# Peer review of "Wake steering optimization under uncertainty"

_Wind Energy Science, 2019_

## Referee Comment (RC1) · Anonymous Referee #1 · 7 Nov 2019

This paper investigates the impact of uncertainty in inflow conditions on the optimization of wake steering strategies in wind plants by formulating a design optimization under uncertainty problem, compared to existing research which typically formulates a deterministic optimization.

The paper is well-written, and different cases are investigated, and the discussions and conclusions are supported by corresponding results.

The findings in this paper have important guidance on the plant control strategies, that is, it is important to incorporate the various uncertainties in plant control analysis.

Overall, acceptance is suggested after the following minor comments are addressed.

1. The x, y labels of figure 1 are not showing up correctly

2. Page 6, line 4, the parameters are assumed to be independent, is this a reasonable assumption? How about dependence between them, how would the dependence impact the results? Some discussions would be helpful.

3. Section 2.3.2, line 8-9, is polynomial chaos expansion used to approximate f_10 for given value of mu, sigma? What are the inputs and output for the polynomial chaos expansion? The size of the training set? A bit more details can be added. Also, some discussion on the computational effort of the numerical model could be helpful to motivate the use of polynomial chaos expansion.

4. Following the earlier comment, unless I am misunderstanding some of the discussions in first paragraph of section 2.3.2, the statement in line 10-11 in section 2.3.2 may not be accurate. Polynomial chaos expansion is some surrogate modeling technique to reduce the computational effort of directly running the expensive numerical model, while quadrature is for integration, they are two different methods for two different purposes, and they cannot be directly compared.

5. Page 8, line 6-8, what is the nested sampling routine? some explanation could be helpful, since this is related to the calculations/simulations that are done.

6. Page 9, line 1, the meaning of this sentence is not clear.

---

## Referee Comment (RC2) · Erik Quaeghebeur (Referee) · 15 Nov 2019

**1 General comments**

**1.1 Summary of key points**

The paper is well written and its topic is relevant. The authors give a good introduction, providing context to problem, and also provide a good overview of literature references.

The authors extend previous work on optimization under uncertainty of wake steering offset angles in a wind farm context. Beyond the already investigated effect of uncertainty in yaw angles, wind inflow direction, they additionally include uncertainty of wind

speed, turbulence intensity, and shear.

The general setup as an optimization-under-uncertainty problem is sensible. The uncertainties considered essentially translate to changing the 'deterministic' wake model into a spread-out 'stochastic' wake model that is obtained by averaging the deterministic one over the probability distributions describing the uncertainties. Their analysis is well structured, dealing first with an exploratory 2-turbine case and then with a more realistic 11-turbine case. The conclusions are (generally, but not always) well supported.

However, I think there is a problem with the treatment of turbulence intensity (TI). The authors treat TI as if it is unrelated to wind speed statistics, whereas they completely determine it. I will discuss this in more detail below, but as a consequence I think the authors will have to remove the entire current treatment of TI and redo their analysis with correct TI values (derived from wind speed statistics).

Furthermore, I think in the uncertainty estimation and effect threshold choice, there is a lack of justification. Again, I will provide more details below. Either these things should be much better justified or the analysis should be redone with values that you can justify.

**1.2  Overview of review aspects**

My judgments here are based on my current understanding of the work.

1. *Does the paper address relevant scientific questions within the scope of WES?*
   **Yes.** Wake steering is relevant for wind farm control. Taking account of the impact of uncertainty makes it possible to define wake steering strategies that are robust against variations captured by the assumed uncertainties.

2. *Does the paper present novel concepts, ideas, tools, or data?*
   **Yes.** It appears to me that the combination of optimization problem definition,

data used, and set of uncertain variables is novel. Each individually is not.

3. *Is the paper of broad international interest?*
**Yes.** It is in principle relevant for any wind farm control strategy designer.

4. *Are clear objectives and/or hypotheses put forward?*
**Not really.** P2L22–29 describes what is going to be done in the paper. This can be reformulated to make it clear what the objectives are.

5. *Are the scientific methods valid and clear outlined to be reproduced?*
**Yes.** The paper contains the necessary information for the study to be reproduced. However, data files for the wind resources and farms used should additionally be supplied.

6. *Are analyses and assumptions valid?*
**Not all.** Assumptions about TI are incorrect.

7. *Are the presented results sufficient to support the interpretations and associated discussion?*
**Almost all.** There is a strange claim about 4% improvement in the conclusions, but that may be a typo.

8. *Is the discussion relevant and backed up?*
**Mostly, but not always.** There is a lack of justification for the uncertainty estimation and effect threshold choice. A few statements need to be better justified, reformulated, or plain removed (see annotated pdf). The discussion should be made more relevant by placing the paper's results in the context of existing results in the literature (I think there are some that are quite similar).

9. *Are accurate conclusions reached based on the presented results and discussion?*
**Yes.**

[Figure]

10. *Do the authors give proper credit to related and relevant work and clearly indicate their own original contribution?*
**Mostly.** Clearly indicating what their own contribution is could be done more explicitly.

11. *Does the title clearly reflect the contents of the paper and is it informative?*
**Mostly.** The title is apt, but quite generic. There is quite some activity in this area and it could be helpful to make the title a bit more specific. (I have no concrete suggestion.)

12. *Does the abstract provide a concise and complete summary, including quantitative results?*
**Yes.**

13. *Is the overall presentation well structured?*
**Yes.**

14. *Is the paper written concisely and to the point?*
**Yes.**

15. *Is the language fluent, precise, and grammatically correct?*
**Yes.** The English is excellent.

16. *Are the figures and tables useful and all necessary?*
**Yes.**

17. *Are mathematical formulae, symbols, abbreviations, and units correctly defined and used according to the author guidelines?*
**Almost all, yes.** $\Sigma$ should be slanted, because it is a variable.

18. *Should any parts of the paper (text, formulae, figures, tables) be clarified, reduced, combined, or eliminated?*

**A few.** Some clarifications may be useful; these are pointed out in the annotated pdf.

19. *Are the number and quality of references appropriate?*
    **Yes.**

20. Is the amount and quality of supplementary material appropriate and of added value?
    **Could be better.** Data for wind resources and farm layouts should be made available and also (other input) and output files could be made available.

**2 Specific comments**

**2.1 Treatment of turbulence intensity**

Turbulence intensity for a given time interval is defined as the ratio of the wind speed standard deviation over the average wind speed for that time interval: $\mu_{\text{TI}} = \sigma_{u_\infty}/\mu_{u_\infty}$ in the notation of the paper. The time interval used can be chosen, but is typically 10 minutes. Whatever the time interval, the definition in terms of wind speed remains.

In the paper, turbulence intensity is introduced as a variable of its own right and even treated as a random variable independent from wind speed. This is incorrect. Turbulence intensity values should be obtained from wind speed statistic values as presented above. I think it makes no sense to treat it separately. So effectively, I think TI should be removed from the list of variables for which uncertainty is assumed and the analysis should be redone with TI values derived from wind speed statistics.

From the estimation of the TI uncertainty you present, P6L15–18, a time variation interpretation of the uncertainty is suggested. However, from the discussion of $f_{10}$ P4L8–19, I did not get the impression that this was the interpretation you wanted to

use. In any case, the FLORIS wake model is quasi-static, which implies that there is some implicit shortest time scale for which it can be applied, as any changes in the parameters that define it would 'in reality' corresponds to a smooth transition between resulting wake profiles. I assumed that this time scale was 10 minutes, but this would be incompatible with the way in which you estimate TI. But even if your assumed time scale is indeed shorter, then still you cannot assign a value to TI independently of the value of $u_\infty$.

**2.2 Uncertainty estimation and effect threshold**

In your uncertainty estimates, there are some strange decisions. Namely, your chosen $\nu_y$ is a bit smaller than the measured one and your wind speed and direction uncertainty are larger than warranted by your sources.

Furthermore, there is absolutely no justification given for the shear parameter uncertainty, but still you call it a worst-case uncertainty. The impression is that you wanted to also include shear in your uncertainty analysis, but did not find any good value for the uncertainty. This should be better justified, or shear should be removed from the uncertainty analysis.

Your choice of threshold for calling the effect of an uncertainty significant (P9L1) is unjustified, currently. This, in combination with the strange decisions may have influenced the picture sketched in Table 4 and P10L4–6. Namely, it might well be possible that if you hadn't made the strange choices, $y$ rather than $u_\infty$ would have been considered significant.

This whole treatment needs to be either much better justified or redone with values that you can justify.

**3   Other comments**

I provide various (other) comments in the annotated pdf of the paper provided in attachment. While not as important as the comments I have worked out in this report, they should also be considered seriously by the authors.

Please also note the supplement to this comment:

[revised manuscript text omitted]

reference shape values. Uncertainties with maximum VSS larger than 0.5% were selected to be included in the wind farm OUU.

The wind farm wake steering optimization problem is intended to provide insights on the benefits of OUU in more realistic scenarios. The plant layout is shown in Figure 4, and the corresponding annual wind speed and direction probability mass function is shown in Figure 5. We performed deterministic and stochastic wake steering optimizations for each speed and direction, reporting the deterministic and expected power production associated with the OUU, deterministic, and baseline strategies. We used the annual wind speed and direction probability mass function to aggregate these speed- and direction-specific power production estimates into an estimate of AEP. The expected power production was maximized during the optimization. The COBYLA optimization driver in DAKOTA (Adams et al., 2014) was used to design the wake steering strategies. The polynomial chaos expansion tool in DAKOTA (Adams et al., 2014) was used during each optimization iteration to estimate the stochastic response in the OUU. Each OUU was initialized with the corresponding deterministic solution.

**3 Results**

[revised manuscript text omitted]

5   uncertainty ($\boldsymbol{\nu}_y = 5°$), but produces a noticeably different solution near $\boldsymbol{\nu}_y = 10°$, which has a maximum VSS of 1.23%. Turbulence intensity affects the wake expansion geometry, which effectively smears out the path of wakes, decreasing the velocity deficit felt by waked turbines. Although we did not find turbulence intensity uncertainty to be significant here, we found a maximum VSS of 1.29% when $\sigma_{\text{TI}} = 10\%$. Such a large standard deviation in this truncated normal distribution approaches a uniform distribution, which could be thought of as representing a complete lack of information regarding turbulence intensity.

[revised manuscript text omitted]

---

## Author Comment (AC1) · 15 Jan 2020

**Response to Referee 1**

We greatly appreciate the time taken by the referee to read our manuscript. We have taken into consideration and addressed all comments, questions, and suggestions from the reviewer, and we feel that the revised manuscript is now substantially stronger as a result. Changes made to the text at the request of the reviewer have been highlighted in red in the revised manuscript. In the following, reviewer comments are repeated in italics and our responses are provided in the bulleted sections of text.

*The x, y labels of figure 1 are not showing up correctly.*

[Figure]

• The axis labels on this figure have been fixed, and we have also added arrows directing the reader to the correct vertical axis for each curve.

*Page 6, line 4: The parameters are assumed to be independent, is this a reasonable assumption? How about dependence between them, how would the dependence impact the results? Some discussions would be helpful.*

• We have added more discussion of the assumption of independence in the Results section. Specifically, on page 7 lines 26-29 of the revised paper, we now note that the uncertainty values assumed will be unique and different for each wind power plant, as several factors influence the ability to measure the relevant inputs to the FLORIS model. We also note that correlations, such as between wind speed and shear, turbulence intensity or yaw error, would change our results. Although we do not consider these correlations in the present study, we now note in the Conclusions on page 18 lines 7-8 that this should be examined in future work.

*Section 2.3.2, line 8-9: Is polynomial chaos expansion used to approximate $f_{10}$ for given value of mu, sigma? What are the inputs and output for the polynomial chaos expansion? The size of the training set? A bit more details can be added. Also, some discussion on the computational effort of the numerical model could be helpful to motivate the use of polynomial chaos expansion.*

• We now note on page 8 lines 4-8 of the revised paper that polynomial chaos expansion was selected to compute $f_{10}$ instead of a simple lower-order quadrature, which would require a very fine-spaced grid of quadrature points or a Monte Carlo approach involving on the order of millions of simulation evaluations. On page 8, lines 13-14 we also now reference the DAKOTA theory manual, which explains the relationship between the number of points sampled, the input dimension, the quadrature order, and the refinement scheme.

*Following the earlier comment, unless I am misunderstanding some of the discussions in first paragraph of section 2.3.2, the statement in line 10-11 in section 2.3.2 may not be accurate. Polynomial chaos expansion is some surrogate modeling technique to reduce the computational effort of directly running the expensive numerical model, while quadrature is for integration, they are two different methods for two different purposes, and they cannot be directly compared.*

• We agree that this is a potentially confusing point. The referee is correct that the polynomial chaos expansion approach fits surrogate models to the more expensive numerical model, however we are integrating this surrogate model in our estimation of $f_{10}$, allowing for fewer quadrature points and better accuracy. We have clarified this starting on page 8 line 4 of the revised paper.

*Page 8, line 6-8: What is the nested sampling routine? Some explanation could be helpful, since this is related to the calculations/simulations that are done.*

• We have now clarified this to note that we are referring to the polynomial chaos nested sampling routine.

*Page 9, line 1: The meaning of this sentence is not clear.*

• We agree that this sentence was potentially confusing, and we have subsequently expanded our discussion of how the uncertain parameters in the 11 turbine wind plant case were selected. As is now noted on page 9 lines 7-9, and starting on line 3 of page 14, we used results from the two-turbine case in order to down-select the number of uncertain parameters considered in the 11 turbine wind plant case. In particular, given the maximum VSS results summarized in Table 4 for the two-turbine case, we chose to consider only uncertainties in the wind inflow direction and speed for the 11-turbine wind plant case. This choice was driven by both the relative impacts of different uncertainties, as well as by the computational cost associated with accounting for each uncertainty. The maximum VSS for $y$ is roughly half that for $u_\infty$, which is itself nearly

an order of magnitude smaller than the maximum VSS for $\theta$. However, consideration of yaw offsets for each turbine in the wind plant case increases the stochastic dimension of the problem by 11, resulting in substantial additional computational expense in the OUU problem. Consequently, we only considered uncertainty in $u_\infty$ and $\theta$ for the wind plant case, with the understanding that the approach is readily extended to include other sources of uncertainty given sufficient computational resources.

Sincerely, the authors.

---

## Author Comment (AC2) · 15 Jan 2020

**Response to Referee 2**

We greatly appreciate the time taken by Erik Quaeghebeur to read our manuscript. We have taken into consideration and addressed all comments, questions, and suggestions from the reviewer, and we feel that the revised manuscript is now substantially stronger as a result. Changes made to the text at the request of the reviewer have been highlighted in red in the revised manuscript. In the following, reviewer comments are repeated in italics and our responses are provided in the bulleted sections of text.

**General Comments:**

*Are clear objectives and/or hypotheses put forward?* **Not really.** *P2L22–29 describes what is going to be done in the paper. This can be reformulated to make it clear what the objectives are.*
• We now explicitly note on page 2 lines 28-29 of the revised paper that our objective is to understand how uncertainty in the examined parameters influences wake steering optimization, and what effect that uncertainty may have on the performance of a hypothetical wind power plant.

*Are the presented results sufficient to support the interpretations and associated discussion?* **Almost all.** *There is a strange claim about 4% improvement in the conclusions, but that may be a typo.*
• We agree that this was potentially confusing and we now note on page 17 lines 18-19 that the 4% refers to the 5 m/s wind speed bin and 0 direction from Figure 9.

*Is the discussion relevant and backed up?* **Mostly, but not always.** *There is a lack of justification for the uncertainty estimation and effect threshold choice. A few statements need to be better justified, reformulated, or plain removed (see annotated pdf). The discussion should be made more relevant by placing the paper's results in the context of existing results in the literature (I think there are some that are quite similar).*
• In our responses to the specific comments from the reviewer (provided below), we note that we now include additional discussion of the uncertainties used in the analysis, and the threshold used to down-select the uncertainties considered in the 11-turbine wind plant case. We also now note the similarity of our results to those obtained by Rott et al. (2018) and Simley et al. (2019) on page 2 lines 32-33 of the revised paper, and explicitly contrast our approach compared to these prior studies on page 17 lines 3-4.

*Do the authors give proper credit to related and relevant work and clearly indicate their own original contribution?* **Mostly.** *Clearly indicating what their own contribution is could be done more explicitly.*

• We agree with the referee, and we have correspondingly added text to the Conclusions beginning on page 17 line 3 noting that, while previous approaches have used simple quadrature to estimate the expectation, we demonstrated the more computationally efficient polynomial chaos approach. We also show that uncertainty in inflow direction effectively smears out the wake. Although the yaw positions are not explicitly considered in the objective function, the OUU formulation leads to smaller yaw offsets, which should generally lead to less-extreme loads. This is also noted on page 3 lines 4-6.

*Does the title clearly reflect the contents of the paper and is it informative?* **Mostly.** *The title is apt, but quite generic. There is quite some activity in this area and it could be helpful to make the title a bit more specific. (I have no concrete suggestion).*

• We appreciate the concern that the title is generic and we debated the formulation of several different possible titles. However, like the reviewer, we were unable to come up with a title that was more appropriate than the current one, and thus we respectfully request that the title be kept as is.

*Are mathematical formulae, symbols, abbreviations, and units correctly defined and used according to the author guidelines?* **Almost all, yes.** $\Sigma$ *should be slanted, because it is a variable.*

• We changed the $\Sigma$ symbol to $\Sigma$. Unfortunately, Latex's bold Sigma is very similar to the bold and italicized Sigma. We also replaced the $\Sigma_i$ symbols with $\Sigma_i$.

*Should any parts of the paper (text, formulae, figures, tables) be clarified, reduced, combined, or eliminated? **A few.** Some clarifications may be useful; these are pointed out in the annotated pdf.*
• We respond to the pdf comments in the responses to specific comments below.

*Is the amount and quality of supplementary material appropriate and of added value? **Could be better.** Data for wind resources and farm layouts should be made available and also (other input) and output files could be made available.*
• We agree that more supplemental material would be helpful for the reader, and we have thus added a representative FLORIS input file, which includes the turbine positions, as well as the wind rose data used for the 11-turbine wind plant test case.

**Specific Comments:**

Treatment of turbulence intensity
*Turbulence intensity for a given time interval is defined as the ratio of the wind speed standard deviation over the average wind speed for that time interval: $\mu_{\text{TI}} = \sigma_{u_\infty}/\mu_{u_\infty}$ in the notation of the paper. The time interval used can be chosen, but is typically 10 minutes. Whatever the time interval, the definition in terms of wind speed remains. In the paper, turbulence intensity is introduced as a variable of its own right and even treated as a random variable independent from wind speed. This is incorrect. Turbulence intensity values should be obtained from wind speed statistic values as presented above. I think it makes no sense to treat it separately. So effectively, I think TI should be removed from the list of variables for which uncertainty is assumed and the analysis should be redone with TI values derived from wind speed statistics.*
• We agree that the description in the original paper was potentially confusing to the reader, indeed leading to the impression that TI should not be considered as an independent uncertain parameter. In several places in Section 2.2 of the revised

manuscript, we now clarify that all inputs to the FLORIS model are themselves assumed to be 10-minute averages, including the turbulence intensity (TI), and the uncertainties are assumed to be in the measurement or calculation of these averages. We also explicitly note on page 4 lines 5-6 and lines 10-11 that these uncertainties may be due to low-frequency variations, spatial variability, and measurement errors. We do, however, agree that the mean wind speed and TI are correlated and, although we have not considered these correlations in our paper, we do note their importance for future study on page 7 lines 28-29 and page 18 lines 7-8.

*From the estimation of the TI uncertainty you present, P6L15—18, a time variation interpretation of the uncertainty is suggested. However, from the discussion of $f_{10}$ P4L8—19, I did not get the impression that this was the interpretation you wanted to use. In any case, the FLORIS wake model is quasi-static, which implies that there is some implicit shortest time scale for which it can be applied, as any changes in the parameters that define it would "in reality" corresponds to a smooth transition between resulting wake profiles. I assumed that this time scale was 10 minutes, but this would be incompatible with the way in which you estimate TI. But even if your assumed time scale is indeed shorter, then still you cannot assign a value to TI independently of the value of $u_\infty$.*

• We again agree with this point and the changes made to Section 2.2 (noted above) should hopefully clarify our interpretation of the uncertainty, temporal variability, and the FLORIS model. As we now explicitly note on page 3 lines 15 and 28, FLORIS is a steady state model, and our formulation treats AEP as a weighted sum from a probability mass function describing the set of mean speeds and directions. This is thus a statistical approach, as opposed to a time series approach.

Uncertainty estimation and effect threshold
*In your uncertainty estimates, there are some strange decisions. Namely, your*

*chosen $\nu_y$ is a bit smaller than the measured one and your wind speed and direction uncertainty are larger than warranted by your sources. Furthermore, there is absolutely no justification given for the shear parameter uncertainty, but still you call it a worst-case uncertainty. The impression is that you wanted to also include shear in your uncertainty analysis, but did not find any good value for the uncertainty. This should be better justified, or shear should be removed from the uncertainty analysis.*

• We now note at the beginning of Section 2.3.1, where we discuss the present uncertainty estimates, that the formulation and demonstration of the OUU analysis approach is not specific to the uncertainty distributions used in the paper, and that the method is equally valid for other choices of these distributions that may represent different real-world conditions and wind plants. The novelty in our paper is thus not in the specific values of the pdf scale parameters, but rather in the formulation and demonstration of the approach, which may be tailored or modified to suit the specific needs of others in the future. As such, we were primarily interested in representative values of the various parameters, rather than exact values for a specific wind plant. This drove the selection of the uncertainties for $\nu_y$, $u_\infty$, and $\theta$, which are close to, but not exactly the same as, previously published values. As we now note on page 7 lines 15-21 of the revised paper, there is very little data available regarding uncertainty in turbulence intensity and the shear parameter, so we specifically chose uncertainties that seemed large to us, perhaps even unrealistically so. However, we then go on to show for the two-turbine case that, even with these large uncertainties, the effect of uncertainties in these parameters on the overall OUU problem is small compared to the effects of uncertainties in $\theta$ and $u_\infty$. Ultimately, however, readers are welcome to implement their own site-specific uncertainty values.

*Your choice of threshold for calling the effect of an uncertainty significant (P9L1) is unjustified, currently. This, in combination with the strange decisions may have influenced the picture sketched in Table 4 and P10L4–6. Namely, it might well be possible that if you hadn't made the strange choices, $y$ rather than $u_\infty$ would have*

*been considered significant. This whole treatment needs to be either much better justified or redone with values that you can justify.*

• This is an important point and we have subsequently expanded our discussion of how the uncertain parameters in the 11 turbine wind plant case were selected. As is now noted on page 9 lines 7-9, and starting on line 3 of page 14, we used results from the two-turbine case, where all uncertainties were considered, in order to down-select the number of uncertain parameters considered in the 11 turbine wind plant case. In particular, given the maximum VSS results summarized in Table 4 for the two-turbine case, we chose to consider only uncertainties in the wind inflow direction and speed for the 11-turbine wind plant case. This choice was driven by both the relative impacts of different uncertainties, as well as by the computational cost associated with accounting for each uncertainty. The maximum VSS for $y$ is roughly half that for $u_\infty$, which is itself nearly an order of magnitude smaller than the maximum VSS for $\theta$. However, consideration of yaw offsets for each turbine in the wind plant case increases the dimension of the stochastic problem by 11, resulting in substantial additional computational expense in the OUU problem. Consequently, we only considered uncertainty in $u_\infty$ and $\theta$ for the wind plant case, with the understanding that the approach is readily extended to include other sources of uncertainty given sufficient computational resources.

**PDF Comments:**
The supplemental document attached by the reviewer made several good points. While some of these points were directly integrated without discussion, we respond to several of the comments in this section.

*P1L13: can you make explicit how this reduces risk? (reduced expected loads? reduced production variance?)*

• The reduced magnitude yaw offsets associated with the OUU optimization reduce

risk of the turbine breaking. We clarified this by adding a citation to Daminani et al. (2018).

*P2L19-21: A comparison of your results with those of these two papers would be appropriate, as the results seem to be quite similar. (Which is good.) This is all the more important given that TI and shear are effectively dropped from consideration after the two-turbine case.*
• We added the following comparison: "Using a polynomial chaos expansion approach, which has not been employed in previous OUU studies of wake steering strategies, we show that direction is the most important uncertain input, effectively smearing out the paths of wakes and reducing the expected velocity deficit. We further show that uncertainty generally reduces prescribed yaw offsets, in agreement with the results of Rott et al. (2018) and Simley et al. (2019) obtained using a simple quadrature approach not based on polynomial chaos expansion."

*P3L6: What is the time scale for which this model is defined? (The wake shape cannot be assumed to change "instantaneously" when the model parameters change, so I feel there must be some minimal appropriate time scale.) I would assume it to be 10 minutes, as it is aimed at AEP calculations as far as I know. If you use it for smaller time scales (I have the impression you do in (2)), I think you should justify that explicitly.*
• The FLORIS model conceives of the inflow conditions as fixed, with the turbine wake in steady state operation. So, there is no intrinsic time scale to the model and we now note on page 3 lines 15 and 28 that the FLORIS model is "steady state". Our approach is thus statistical by nature, meaning there is no transition in time between different inflow or operating conditions.

*P4L4: Is it possible to give the constitutive expressions for this model? (Dependence*

*of C, $\delta$, $\sigma_y$, $\sigma_z$ on the components of v at the very least.)*
• Because the Gaussian wake model is complex, with several tuned coefficients, we now refer the reader to the FLORIS documentation for additional detail on page 3 lines 21-22 of the revised paper.

*P4L9: Can you say which types of uncertainty affects which components of $\nu$ and how? (Make this statement a bit more concrete.) Also, I would think there is epistemic uncertainty beyond measurement error, but perhaps that is not modeled. (Perhaps this is clarified below somewhere.)*
• To maintain clarity in the text, we deleted the sentence in question, but have added text on page 4 lines 5-6 outlining potential sources of uncertainty.

*P4L26: If you are going to leave $\Sigma$ here in totality on the lhs, then it must be made clear at its introduction that it does not contain variability information for $\theta$ and $u_\infty$. Currently there is too much handwaving going on about $\Sigma$.*
• In the formulation of Eq. (4), $\Sigma$ does still contain variability information for $\theta$ and $u_\infty$. This is nested within the calculation of $f_{10}$ from Eq. (3), which is in the integrand of Eq. (4). Through the additional integration in Eq. (4), we thus only remove the dependence on $\mu_{u_\infty}$ and $\mu_\theta$.

*Table 1 You have not introduced this notation with square brackets and two arguments. (You should.) In (4) you define notation with parentheses and four arguments. I'll assume this notation is the same, but with the second and third argument kept the same and therefore elided.*

• The square brackets were a stylistic choice based on the nested ordering of parenthetic symbols according to () first, followed by [], and then {}. As noted in the caption, two of the parameters have been suppressed for clarity.
*P5L17-18: What is the sampling frequency of this data?*
• The probability distribution is computed using 30-second averages of different instrument measurements, resulting in samples every 30 s. This is now noted in the caption of Figure 2.

*P6L4: No, $\nu$ is a scale parameter.*
• The reviewer is correct that this is a scale parameter and not a shape parameter. We revised the wording throughout the document to avoid referring to scale parameters ($\sigma$ or $\nu$) as shape parameters.

*P6L10-11: Why not set it to 6.16(or perhaps 6, rounded) as 5represents a smaller yaw misalignment than actually observed in reality.*
• We decided to choose round numbers as our reference uncertainty values. Readers are welcome to implement site-specific values.

*P6L15: Why not 3—4and 0.5 m/s? In this case you take larger values without explanation. In the MMIJ dataset, for 89 m, I find the average 10-minute $\sigma_\theta$ to be 3.7and the average 10-minute $\sigma_{u_\infty}$ to be 0.6 m/s. A dataset like this, or the OWEZ and FINO ones, allow you to create joint distributions for ($\mu_{u_\infty}$, $\sigma_{u_\infty}$, $\mu_\theta$, $\sigma_\theta$) instead of just for ($\mu_{u_\infty}$, $\mu_\theta$) in (4). That would seem to be a better way than to assume constant values for $\sigma_{u_\infty}$ and $\sigma_\theta$. It would at the same time allow you to get the correct TI for each 10-minute interval, instead of using a likely inconsistent constant $\mu_{TI}$ as you do now.*
• We appreciate the investigation the reviewer did in estimating these uncertainty values. As discussed above, we chose round numbers for simplicity and readers are welcome to implement their own uncertainty values using our methodology.

*P6L19: In what way is [$\sigma_\alpha$] estimated? Formulated as it is now, it just appears posited without justification.*
• There is not a bounty of data available for uncertainty in mean shear over short time periods. In addition, it will be site-specific. Based on our experience, we chose $\sigma_\alpha$ as 0.05 for (what we conceive of as) a very large value.

*P7L2: [Physical variability and aleatoric uncertainty] are essentially the same, no? (Does one need to be replaced by "measurement uncertainty"?)*
• We agree and to avoid confusion we have deleted this sentence.

*P7L8: Some more information showing good convergence behavior for your application would be welcome. In my experience, PCE often has convergence issues. (Although the use of independent standard distributions may make this less of an issue.)*
• We internally performed convergence analysis. The smooth signal of expected power versus inflow direction in Figures 6, 7, and 8 indicate convergence for these cases. When the approximation is not converged, these quantities appear bumpy. The constancy between the lower-order and higher-order expansions brings confidence in our plant OUU results. For these reasons, we feel the convergence of the expansion is clear.

*P7L10-11: But wasn't that for calculating Eq. (4), which is different from Eq. (10)? Put differently: is this reference relevant here?*
• We feel that this reference is relevant since it also uses polynomial chaos expansion (PCE) to estimate AEP. While we use rectangular quadrature to estimate AEP given $f_{10}$, we use PCE to estimate $f_{10}$, which goes into the AEP calculation.

*P7L13: Make it explicit what this means. Is this fifth order for each variable or for their product?*
• We now note that we are referring to the product of the variables.

*P8L3: It looks to be coming from the East. . . I realize this is just a matter of rotation that is irrelevant here. So I would just not even mention this, as it remains jarring to see a North wind come from the left.*
• This geometry fits much better in a page than having the inflow be from the top. Our direction places 0 inflow direction as coming from the North. To avoid confusion in the later plots regarding the two-turbine analysis results, we felt the need to clarify that the direction is from the North.

*Figure 5: Why only [-60,60] and not all directions [-180,180]? (I think a justification should be easy to give, but don't let people guess.) You appear to be using 10direction bins. This is too large for wind farm layout optimization ($< 5$needed). Can you justify its use in your use case of wake angle optimization? At the very least, the resulting wake steering strategy should be evaluated with far smaller direction bins, i.e., of about 1, even if optimization is still done with 10 bins. I guess this will also force you to explicitly state how you would interpolate it.*
• We acknowledge that we chose a relatively limited range of inflow directions as well as relatively large direction bins. Both of these choices were intended to simplify the formulation and demonstration of the OUU problem. Although 10 degree binning is certainly too large for evaluating AEP for the purposes of layout optimization, we think that the coarse binning should not meaningfully influence our comparisons between direction-specific control strategies. To further explain these choices, we now include additional text on page 10 lines 7-11 of the revised paper.

*P10L13-14: This plot should be antisymmetric because of the problem formulation, no? It's almost, but not entirely antisymmetric. Can you explain why? If it should indeed be antisymmetric, a better plot would be to just consider positive angles.*
• We agree that the figure should be antisymmetric and the small differences are due to the slightly different binning of the wind direction near 0. We would respectfully like to leave this figure as-is, precisely because it confirms readers' intuition of antisymmetric results.

*I do not understand why the maximum is away from 0 for higher uncertainty values (and the value at 0 even goes to zero). Can you explain? (One reason why I'm surprised by this is, is that for the case with uncertainty, you are essentially considering a similar wake model, but now wider, so a qualitative change in behavior is unexpected from this viewpoint.)*
• In Figure 6(d), the optimal yaw, which is shown on the vertical axis, actually moves towards zero for larger uncertainty values, which seems to be consistent with the expectation of the reviewer.

*Figure 7: Again, because of the symmetry, I'd just plot for positive inflow angles.*
• We feel that plotting the full range of inflow directions better highlights that there is a discontinuity in this function and helps the reader connect these results to the previous plots. As such, we would respectfully like to leave this figure as-is.

*P12L8-P13L1: Does this effect stay as big when computing AEP using 1 direction bins instead of 10 ones?*
• The trends we discuss are not direction-specific. While we anticipate that the AEP would be different if computed with a smaller bin size, moving to a finer directional bin spacing complicates our demonstration problem, as noted in our response above.

*P14L27-28: I do not understand this: $\mu_y$ appears in both (5) and (6), no?*
• We meant to say that we did not include minimization of the yaw positions as part of the optimization objective, but we agree that this sentence is confusing and we have therefore deleted it.

*P15L2-3: Say why you think [typical levels of uncertainty in turbine power and thrust curves probably would have resulted in somewhat different optimum wake steering strategies]; don't just state this.*
• On page 18 lines 4-5 of the revised document, we now explain that uncertainty in power would change the shape of the mean power curve, altering the tradeoff point, and uncertainty in thrust would affect the magnitude of the velocity deficit behind the turbine.

Sincerely, the authors.